# A Combined Computational and Experimental Analysis of PLA and PCL Hybrid Nanocomposites 3D Printed Scaffolds for Bone Regeneration

**DOI:** 10.3390/biomedicines12020261

**Published:** 2024-01-24

**Authors:** Spyros V. Kallivokas, Lykourgos C. Kontaxis, Spyridon Psarras, Maria Roumpi, Ourania Ntousi, Iοannis Kakkos, Despina Deligianni, George K. Matsopoulos, Dimitrios I. Fotiadis, Vassilis Kostopoulos

**Affiliations:** 1Biomedical Engineering Laboratory, School of Electrical and Computer Engineering, National Technical University of Athens, 15773 Athens, Greece; 2Computation-Based Science and Technology Research Center, The Cyprus Institute, 2121 Nicosia, Cyprus; 3Department of Mechanical Engineering and Aeronautics, University of Patras, 26504 Patras, Greece; 4Unit of Medical Technology and Intelligent Information Systems, Department of Materials Science and Engineering, University of Ioannina, 45110 Ioannina, Greece

**Keywords:** scaffolds, mechanical analysis, fluid flow simulation, bone regeneration

## Abstract

A combined computational and experimental study of 3D-printed scaffolds made from hybrid nanocomposite materials for potential applications in bone tissue engineering is presented. Polycaprolactone (PCL) and polylactic acid (PLA), enhanced with chitosan (CS) and multiwalled carbon nanotubes (MWCNTs), were investigated in respect of their mechanical characteristics and responses in fluidic environments. A novel scaffold geometry was designed, considering the requirements of cellular proliferation and mechanical properties. Specimens with the same dimensions and porosity of 45% were studied to fully describe and understand the yielding behavior. Mechanical testing indicated higher apparent moduli in the PLA-based scaffolds, while compressive strength decreased with CS/MWCNTs reinforcement due to nanoscale challenges in 3D printing. Mechanical modeling revealed lower stresses in the PLA scaffolds, attributed to the molecular mass of the filler. Despite modeling challenges, adjustments improved simulation accuracy, aligning well with experimental values. Material and reinforcement choices significantly influenced responses to mechanical loads, emphasizing optimal structural robustness. Computational fluid dynamics emphasized the significance of scaffold permeability and wall shear stress in influencing bone tissue growth. For an inlet velocity of 0.1 mm/s, the permeability value was estimated at 4.41 × 10^−9^ m^2^, which is in the acceptable range close to human natural bone permeability. The average wall shear stress (WSS) value that indicates the mechanical stimuli produced by cells was calculated to be 2.48 mPa, which is within the range of the reported literature values for promoting a higher proliferation rate and improving osteogenic differentiation. Overall, a holistic approach was utilized to achieve a delicate balance between structural robustness and optimal fluidic conditions, in order to enhance the overall performance of scaffolds in tissue engineering applications.

## 1. Introduction

Biocomposite scaffolds are key players in regenerative medicine applications through mechanical support for tissue and bone regeneration. Three-dimensional (3D) scaffolds are critical in bone tissue engineering, as they promote the regeneration of injured or depleted bone tissue, representing a significant advancement in orthopedic surgery [1]. These scaffolds, meticulously designed and printed using biocompatible materials, mimic the complex structure of natural bone, providing a conducive environment for bone growth and healing. Furthermore, the porous architecture of these scaffolds facilitates vascularization and cell migration, essential for effective bone regeneration [2]. As such, the advent of additive manufacturing through 3D bioprinting has promoted research and development and is strongly related to the rising demand for computational design for scaffolds [3]. Biodegradable and biocompatible polymers such as PLA and PCL are promising materials for these applications [4]. It is important to understand that microstructure plays a vital role in the final efficiency of these materials for such applications. Therefore, a thorough study of their structural, mechanical, and fluid flow-induced properties is necessary to propose new scaffold candidates for bone tissue engineering applications.

The utilization of computational methods has been instrumental in streamlining the design process, minimizing expenses, and tailoring the scaffolds to possess the intended properties. On this premise, finite element analysis (FEA) has been assessed as a valuable tool in structural mechanics and computational fluid dynamics (CFD) analysis for providing a synergistic understanding of both the structural and the fluidic aspects of scaffolds. The simulation and prediction of the mechanical responses of the scaffold provide insights into the scaffold’s behavior under various loading conditions. These simulations consider factors like material composition, pore size, and overall scaffold architecture to evaluate parameters such as stiffness, strength, and elasticity, while allowing for the optimization of scaffold designs before actual fabrication, reducing the risk of mechanical failure and ensuring biocompatibility. By simulating fluid flow through the scaffold architecture, valuable insights into how different design elements impact permeability and wall shear stress (WSS) can be gained, affecting the scaffold’s efficacy in supporting cellular activities. Permeability is a pivotal property governing the transport of nutrients and gases within the scaffold matrix, while WSS indicates the mechanical stimuli experienced by cells within the scaffold environment [5,6]

Numerous studies have been published focusing on biocomposite scaffolds with various materials and geometries to support tissue and bone regeneration [7,8,9,10,11,12,13]. Kakarla et al. [8] used the representative volume elements (RVE) method and FEA to study the maximum stress distributions and mechanical properties of boron nitride nanotubes (BNNTs)-reinforced gelatin (G) and alginate (A) hydrogel. Also, efforts have been made using matrices PLA and PCL with fillers like multiwall carbon nanotubes (MWCNTs), chitosan (CS) and silk [14,15,16,17]. It has been found recently that PCL/CS/CNT composite matches well with the heart’s electrical conductivity [16]. Hendrikson et al. [18] evaluated four scaffold designs with different architectures using additive manufacturing, FEA, and CFD to investigate the influence of additive-manufactured scaffold architecture on the distribution of surface strains and fluid flow shear stresses within the scaffold. The results show that the scaffold architecture affects surface strains and fluid flow shear stresses under mechanical compression and imposed fluid flow, since the various WSS ranges were only acquired by modifying the scaffold architecture. Moreover, Ouyang et al. [10] linked the hydromechanical properties with the pore size of titanium-based scaffolds, indicating the effect of pore size on cell proliferation and bone growth. In the study by Deng et al. [12], titanium-based biocomposites with a porosity of 65% and a number of scaffolds with different geometries were used. The combination of chitosan with biopolymers and clay nanotubes has been explored for the fabrication of composite materials useful for tissue engineering [19]. CFD analysis was employed to calculate the permeability, velocity, and flow trajectory within the scaffold structure. The diamond lattice unit (DIA) structure exhibited the smallest internal fluid velocity difference, and the fluid flow trajectory inside the scaffold was found to be the longest. This characteristic is advantageous for promoting blood vessel growth, facilitating nutrient transport, and fostering bone formation. Another study by Wang et al. [20] investigated the performance of a honeycomb scaffold structure, through a static compression test and CFD analysis to measure the permeability of the scaffolds and their match with the parameters of human bone tissue. The examination and analysis of four sets of honeycomb structures indicate the feasibility and significant research potential of utilizing honeycomb structures as biomimetic bone scaffold structures. Recently, Mohol et al. [21] studied the mechanical behavior through static structural analysis, and they also investigated the fluid dynamics performance and the degradation impact of polylactic acid scaffolds with nature-inspired design structures visualizing the velocity and pressure contours. In order to validate the CFD analysis results, the computed permeability values of all scaffolds in this study were from 5.45 × 10^−7^ to 8.06 × 10^−7^ m^2^, aligning with the permeability of cancellous bone, which falls within the range 10^−8^–10^−7^ m^2^.

The complex and dynamic nature of living tissues provides a range of variables that affect both scaffold mechanics and fluid flow dynamics. Despite substantial efforts and significant research into scaffold mechanical characteristics and the principles governing fluid flow, the wide variety of materials used to fabricate scaffolds and the different scaffold designs further contribute to this multifaceted challenge. To achieve balance in these parameters, the current work proposes a combined computational and experimental approach to determine the effect of a 3D-printed rectangular-shaped scaffold architecture, made from hybrid nanocomposites of PCL and PLA matrices (enhanced with CS and MWCNTs), on mechanical properties and fluid flow dynamics, for potential applications in bone tissue engineering.

## 2. Materials and Methods

### 2.1. Experimental Work

The experimental work involved the manufacturing and mechanical testing of PCL and PLA nanocomposites reinforced with CS-functionalized MWCNTs.

#### 2.1.1. Materials

The MWCNTs were acquired from Nanografi Nanotechnology (Ankara, Turkey), with specific properties (i.e., having an outside diameter of 48–78 nm and purity of more than 96%), while low- and medium-molecular-weight chitosan was purchased from Glentham (Corsham, UK). The PLA and PCL synthetic polymers used as matrices for the functionalized MWCNTs were purchased from 3devo B.V. (Utrecht, The Netherlands) and Thermo Scientific (Waltham, MA, USA), respectively.

#### 2.1.2. Fabrication of the PLA/PCL Hybrid Scaffolds

To functionalize the MWCNTs, two types of chitosan with different molecular weights were used. For low-molecular-weight chitosan (250,000 avg), the MWCNTs were incorporated at a weight fraction of 40%, resulting in a product designated as L40/100. For medium-molecular-weight chitosan (1,250,000 avg), the weight fraction of the MWCNTs was 40%, and the product was labelled as M40/100. The PLA and PCL pellets were pulverized and subsequently thoroughly mixed with the functionalized CS/MWCNTs at a weight fraction of 5%. The goal was to achieve the maximum filler content, while ensuring that the reinforced materials remained printable and did not negatively affect the initial mechanical properties. It was found in preliminary work that a weight fraction of 5% was the maximum attained weight filler fraction at which the materials remained printable. The resulting nanocomposite mixes were designated as (PCL_LNG, PLA_LNG) and (PCL_MNG, PLA_MNG), depending upon the type of chitosan-functionalized MWCNTs used [22].

The PLA and PLC CS/MWCNTs materials were placed in a dehumidifier to remove all moisture. Then the materials were extruded in the form of 3D-printing filament. The filament was once more pelletized and pulverized and was then extruded in the form of filament or the form of bulk materials specimens. The pulverization process was repeated twice, in order to break any existing large agglomerations and achieve a better dispersion of the filler material inside the PLA and PCL matrices. Bulk materials and scaffolds were manufactured using 3D printing (printer 3DISON AEP by ROKIT, Seoul, Republic of Korea). In our previous work, cubic-shaped 3D-printed scaffolds were manufactured, with dimensions 5 × 5 × 5 mm^3^ [23]. However, this resulted in specimens with imperfections due to the small size. Therefore, we decided to manufacture specimens of 15 × 15 × 5 mm^3^ and subsequently cut 9 cubic specimens of 5 × 5 × 5 mm^3^ using a microtome. Scaffolds made from different materials had the same dimensions and geometry. Figure 1 shows an example of a reinforced PCL scaffold. The photo was taken with a microscope (Optika szm-Led2 by OPTIKA, Bergamo, Italy).

#### 2.1.3. Mechanical Testing

To determine the mechanical properties of the specimens, compression tests were performed. Specifically, both pure and reinforced PCL/PLA underwent compression tests, at a constant crosshead speed of 2 mm/min, in accordance with ASTM D0695, using a universal testing machine (INSTRON 8872, High Wycombe, UK), and at ambient temperature. For the bulk materials, the compressive moduli, the apparent compressive moduli, and the compressive strength were calculated, while for the 3D-printed scaffolds, the force versus displacement curves were calculated.

### 2.2. Computational Work

In this study, the computational work incorporates two district simulations: a structural analysis, to predict the mechanical behavior of the scaffolds, and a CFD analysis, to predict how scaffold geometry architecture influences fluidic parameters such as permeability and WSS.

#### 2.2.1. Mechanical Properties

The structural analysis was implemented utilizing FEA and employing Abaqus 2023 software to gain insights into the behavior of porous scaffolds at the microscale under controlled compressive displacement. The FEA focused on the given rectangular geometry with dimensions of 5 × 5 × 5 mm^3^ and 45% porosity, consistent with the experimental setup. Figure 2 provides a visual representation of the geometry of the 3D scaffold used in the analysis. To ensure accurate and reliable results, prior to the mechanical modeling, a mesh sensitivity study was conducted [24]. For this study, a specific displacement load was applied while using grids with different resolutions and estimating the level of alteration of the converged solution with each mesh created. Specifically, the element dimensions gradually decreased from an initial length of 1 mm to 0.1 mm in a systematic manner by applying a displacement of up to 1 mm. Following this convergence test, an element size of 0.25 mm was selected for the analysis. Throughout this process, the brick-type element C3D8R was used.

Following the mesh sensitivity analysis, a comprehensive approach was implemented, aimed at understanding the stress distributions within the scaffold structure. The same scaffold design was employed, with materials being considered homogeneous, isotropic, and elastic–plastic. The parameters for the different materials (PCL_LNG, PLA_LNG, PCL_MNG, PLA_MNG) were extracted using Abaqus software, giving as input parameters the stress–strain curves of the bulk/reference materials. Poisson ratios for the PLA and the PCL (0.3 and 0.4, respectively) were acquired from relevant studies [4].

To assess the mechanical properties, a compressive displacement was applied to all systems along the y-axis, mimicking the experimental conditions. The displacement was directed opposite to the positive y-axis, and the xz plane at the bottom layer of the scaffold was fixed. In addition, to enhance the fidelity of the simulation outcomes and further replicate the compressive experiment conditions, a rigid planar surface was introduced at the top loading face of the porous scaffold cube during the simulation, while the bottom faces of the scaffold were fixed. A reference point was also defined, indicating the location where the displacement was applied (Figure 2). This setup aimed to closely emulate the real-world experimental conditions.

#### 2.2.2. Governing Equations and Boundary Conditions

For the CFD analysis, the Navier–Stokes and continuity equations were employed to investigate laminar flow of an incompressible fluid characterized by constant density and viscosity:(1)$$\rho \frac{\partial u}{\partial t}-\mu \nabla^{2}u+\rho \left(u\cdot \nabla \right)u+\nabla p=0,\nabla \cdot u=0,$$
where ρ is the density (kg/m^3^), u is the velocity (m/s), µ is the dynamic viscosity of the fluid (kg/(ms)), and p is the pressure (Pa). The fluid properties of the cell culture medium were assigned with a density of 1008 kg/m^3^ and a viscosity of 0.0078 kg/m/s [25,26]. No-slip no-penetration was applied on the surface of the scaffold, and an outside fluid domain was created to avoid boundary effects. Three different inlet velocities (0.05 mm/s, 0.1 mm/s, and 0.5 mm/s) were investigated, while the outlet pressure was assumed to be zero [25,26] (Figure 3).

The scaffold’s permeability (K) using Darcy’s law was:(2)$$K=\frac{v_{inlet}\mu ∆h}{∆P},$$
where $v_{inlet}$ is the inlet fluid flow velocity (mm/s), μ is the dynamic fluid viscosity (kg/ms)), ΔP is the pressure drop (Pa), and ∆h  is the distance (mm) from the inlet to the outlet surface [27].

The WSS ($\tau_{w}$) was computed as:(3)$$\tau_{w}=\mu \frac{\partial u}{\partial n}\;,$$
where n indicates the normal direction to the corresponding plane.

#### 2.2.3. Sensitivity Analysis

A mesh sensitivity analysis was conducted in a representative case, assuming a rigid wall with steady-state flow. The sensitivity analysis for the no-slip, no-penetration wall boundary condition involved simulations with a face size ranging from 5 to 2 mm. The analysis focused on establishing the correlation between mesh size and the resulting average values for pressure, velocity, and WSS. Mesh sizes exhibiting less than a 5% difference in parameter values were selected for the simulations. The mesh size outside the scaffold region was gradually reduced from 5 to 2 mm. The three-dimensional meshing process was standardized across all cases once the mesh was generated. Tetrahedral elements were used for the model. We started the discretization using an element face size of 5 mm, which corresponds to 2,909,644 million elements, reducing to a mesh face element size of 2 mm, which corresponds to 20.92 million elements.

## 3. Results

### 3.1. Experimental Results

Compression experiments were carried out for all bulk and scaffolds materials, with the test samples being manufactured using 3D printing. Table 1 provides the experimental values for the compression modulus of the bulk materials, while Figure 4 illustrates the experimental apparent compressive moduli and ultimate compressive strength of all tested scaffolds.

It was observed from Table 1 that for the bulk material tests the PCL’s modulus was not affected significantly by the reinforcement. The PCL reinforced with the medium-molecular-weight chitosan (PCL_MNG) possessed a slightly higher compressive modulus when compared with the pure PCL. Similarly, the PCL reinforced with the low-molecular-weight chitosan (PCL_LNG) displayed a higher compressive modulus than the pure PCL. The PLA specimens exhibited similar behavior when reinforced with the CS/MWCNTs. A small increase in the specimens reinforced with the medium-molecular-weight chitosan (PCL_MNG) was observed; however, the difference was only marginal. It should be noted that reinforcing the scaffold with different materials did not aim to significantly change the mechanical properties but aimed to promote and improve cell proliferation and differentiation. In this regard, although the difference in compression modulus between the PCL and the PLA was significant, this fact did not directly translate into the scaffolds’ behavior, which is evident in Figure 4a. For the compressive experiments implemented for the scaffolds, the PLA scaffolds still exhibited higher apparent moduli compared to the PCL ones. Interestingly, the substantial difference in the bulk/reference material compression moduli between the PCL and the PLA specimens (as depicted in Table 1) was not mirrored in the scaffolds’ apparent moduli. Within the scaffold specimens, the apparent modulus for the PCL and the PLA-based scaffolds was of the same order of magnitude, while for the bulk PLA it was roughly nine times larger than for the bulk PCL. This indicates that the geometry and the manufacturing method of the scaffolds influence the mechanical behavior of the scaffolds more than the bulk material properties. Furthermore, the incorporation of CS/MWCNTs reinforcement did not lead to significant fluctuations in the apparent compressive modulus. For instance, in the PCL-reinforced scaffolds, the apparent compressive modulus decreased. On the other hand, in the PLA scaffolds, as well as the PLA_LNG and the PLA_MNG, the addition of CS/MWCNTs did not significantly increase the apparent compressive modulus, and the difference can be considered negligible, and within the experimental error margins. Regarding the ultimate compressive strength (Figure 4b), a noteworthy reduction in strength was observed, which can be attributed to issues introduced during manufacturing, discussed in Section 4.

### 3.2. Computational Results

#### 3.2.1. Mechanical Simulation

As previously mentioned, the material behavior was assessed using Abaqus software, providing valuable insights into the mechanical behavior of the scaffolds and the way the different materials affect stress distribution and deformation. Subsequently, the distributions of von Mises stresses and the force versus displacement curves along the structure of the different scaffolds were calculated. Figure 5 shows the stress distribution along the microstructure of the scaffold made from PLA and PCL polymers, with the given geometry, compressed under controlled displacement. Figure 6 and Figure 7 present the distribution of von Mises stresses at the middle cross-section for the PCL and PLA scaffolds, respectively, as well as for their respective CS/MWCNT-reinforced counterparts.

The simulation indicated that, in all cases, the stresses were higher in the cross-sectional areas of the system during normal compressive tests. Upon further inspection, it is evident that the PLA scaffolds had higher endurance for the same displacements, both at the surface level and inside the scaffolds, with the PLA_LNG yielding the optimal results.

Following the stress assessment, the force versus displacement curves were estimated, taking into account both the experimental and computational analysis for all types of scaffolds. Figure 8 displays the force–displacement curves of the PLA-based scaffolds. Overall, there is very good agreement between the experimental results and the FEM analysis, except for the scaffold made of pure PLA material. In this case, a notable discrepancy between experimental and FEM results appears in the displacement at forces from 180 N to 350 N. A similar, though minor, inconsistency is evident in the case of the PLA_LNG samples. This may be due to impurities that were induced during the manufacturing process. As we can see, these two systems, in contrast with the PLA_MNG, are experiencing a smoother transition to the plastic region. In this direction, impurities can work in such a way that they intensify the transition between the two regions.

The force–displacement curves for the PCL-based scaffolds are demonstrated in Figure 9. Although there is a good agreement between the experimental results and the FEM analysis, in the case of the PCL_MNG, the FEM analysis predicts a lower maximum load and earlier collapse of the scaffold structure.

#### 3.2.2. Fluid Flow Dynamics within Scaffolds

Using ANSYS 16.2 software, CFD analysis was performed to investigate how the scaffold design can affect fluid flow patterns within scaffolds and optimize the fluidic conditions. The same scaffold geometry was used as in the structural analysis (Figure 2). The parameters of velocity, WSS, and permeability of the scaffold were analyzed at three different inlet velocities of 0.05 mm/s, 0.1 mm/s, and 0.5 mm/s.

The velocity streamline distribution at different inlet velocities is demonstrated in Figure 10. The fluid flows through all the struts in the scaffold geometry. Figure 11 demonstrates the distribution of velocity streamlines at the middle cross-section for the scaffold geometry. The higher velocity in the struts, as depicted in Figure 11c, indicates the increase in the rate that is favorable for the absorption of cells and nutrients on the inner surface of the scaffold. This accelerated velocity creates an environment conducive to more efficient cellular interactions and improved nutrient uptake, ultimately contributing to the overall effectiveness of the scaffold in supporting biological processes [26].

##### Permeability

The findings from the permeability analysis of the proposed scaffold design (Figure 2) were compared to those derived by other studies that use rectangular-shaped bone scaffolds [12,26,28]. The comparison of the permeability findings for the scaffolds is prsented in Figure 12.

The permeability values for human natural bone varied from 10^−11^ m^2^ to 10^−8^ m^2^, thus the results in the current work are in good agreement with the literature data [29]. The three different inlet velocities did not produce significant changes in the value of the scaffold’s permeability, as depicted in Table 2.

#### 3.2.3. Wall Shear Stress

In the present study, the distribution of WSS in the scaffold is demonstrated in Figure 13a. WSS contours within the scaffold indicate that its magnitude in different areas varied from zero to a few mPa. The maximum WSS occurred in areas close to the inlet. Regarding the measured value of average WSS, this was calculated to be 2.48 × 10^−3^ Pa, which is within the range of the reported literature values for promoting osteogenesis [30]. The average WSS values obtained for the scaffold geometry at different inlet velocities are demonstrated in Figure 13b. The CFD results revealed that WSS and fluid flow rate have a linear relationship and consequently any decrease in velocity leads to a reduced WSS value.

## 4. Discussion

In this paper, an experimental and computational assessment was performed with regards to PLA and PCL reinforced with CS/MWCNTs scaffolds with a rectangular-shaped scaffold design. The suitability for PLA scaffold fabrication for bone reconstruction applications is further supported due to its accelerated bone regeneration properties [9,13,21]. It should be taken into consideration that the material selected for scaffold design should also address properties (apart from mechanical strength) such as biocompatibility and degradation behavior. The addition of CS provides versatility in scaffold design and customization offers advantages in tissue engineering [16,19,22].

For the experimental results, the mechanical testing implemented on the reinforced PCL and PLA bulk materials resulted in minor changes compared to the pure PCL/PLA compression modulus (Table 1). This can be attributed to the reinforcement used, which limits the movement of the molecular chains and increases the compression modulus since MWCNTs are acting in general as nucleating agents and could increase the crystallinity of PLA, although the increase is marginal because of the small amount of MWCNTs and their functionalization with CS [31,32]. Finally, since longer polymer chains suggest a larger entanglement of the polymer chains, leading to a higher compressive modulus, PCL_MNG offers better reinforcement than PCL_LNG. However, the differences are marginal and well within experimental error. Interestingly, the PLA_LNG presented a lower compression modulus compared to pure PLA and the PLA_MNG. However, the difference is ~1%, and thus no conclusion can be drawn. Furthermore, concerning the mechanical properties of the scaffolds (Figure 4), the PLA-based scaffolds demonstrated higher apparent moduli compared to the PCL-based ones. Notably, the apparent compressive modulus decreased in the PCL-reinforced scaffolds when incorporating CS/MWCNTs reinforcement, while increasing in the reinforced PLA CS/MWCNTs. However, the changes in all materials were not significant and were within the margin of error. On the contrary, the compressive strength of all six scaffolds presented a reduction from pure (PLA or PLC) to CS/MWCNTs reinforcement. Generally, well-bonded fillers contribute to composites having notably higher strength, while weakly bonded particles can act as sources of inherent flaws, leading to a decrease in strength [33]. In the case of nanoscale reinforcement, nanofillers tend to agglomerate more, creating crack-provoking flaws and reducing strength [34,35]. This study observed that the addition of MWCNTs, especially in the PCL-loaded scaffolds led to nozzle clogging during the printing process, and the 3D-printed nanocomposite exhibited increased brittleness when compared to the pure PCL scaffolds. This was to be expected, since our goal was to achieve the maximum filler content, while ensuring that the reinforced materials remained printable and did not negatively affect the initial mechanical properties, in order to promote and improve cell proliferation and differentiation. The aforementioned factors, despite being addressed to the best of our abilities during manufacturing, could explain the overall reduction in ultimate compression strength observed.

Concerning the mechanical modeling, PLA scaffolds demonstrated higher endurance for the same displacements, both at the surface level and inside the scaffolds. This can be attributed to the lower molecular mass of the PCL_LNG filler compared to the PCL_MNG, providing more flexibility. The comparison between the PCL and the PLA-based scaffolds also revealed that the PLA scaffolds exhibited higher reaction forces, resulting in higher moduli and ultimate strengths. Among the PLA variations, the PLA_MNG demonstrated a higher elastic modulus, while the PLA_LNG provided a favorable combination of elastic modulus and ultimate strength. These observations highlight the complexity and challenges in accurately modeling the mechanical behavior of such intricate structures, especially when dealing with variations in material properties and microstructural features. Adjustments to the material model and consideration of microbuckling effects could enhance the simulation accuracy. The subsequent force versus displacement estimation also verified our simulation, with the experimental and simulation values being in good agreement (Figure 8 and Figure 9). The small divergence in the values of the two approaches might be attributed to the microbuckling of the scaffold elements at these loads.

Taking these findings into consideration, it can be inferred that the choice of material (PLA) with a specific type of reinforcement (MNG or LNG) can significantly influence the material’s response to mechanical loads, leading to an optimal structural robustness.

Computational fluid dynamics analysis offers a synergistic framework for a deeper understanding of the scaffold’s fluidic dynamics, taking into account the scaffold’s architecture. The flow rate of the fluid (cell culture medium) in the internal structure of the scaffold significantly affects the growth of bone tissue in the scaffold, with an increased flow rate being favorable for nutrient transport and cellular response. Also, the parameter of the scaffold’s permeability, which determines the material’s ability to allow fluid flow, was identified as a significant factor that influences the growth of bone tissue [36]. Permeability promotes cell attachment to the scaffold surface, so a lower value of permeability cannot deliver adequate nutrients. In this study, the permeability of the scaffold is in the acceptable range close to that of human natural bone permeability (Figure 11), influencing osteogenesis and vascularization in bone regeneration. Recent studies [12,26,27] cite findings that scaffolds with higher permeability enhanced bone formation both in vitro and in vivo.

Moreover, WSS plays a major role in stimulating cell proliferation within scaffolds. Studies of the literature indicate that WSS within the range of 0–30 mPa enhances the overall biological activity of mesenchymal stromal cells (MSCs). In the range of 0.11–10 mPa, it specifically promotes osteogenic differentiation, while in the slightly broader range of 0.55–24 mPa, it stimulates the mineralization process of bone cells. However, WSS values exceeding 60 mPa are associated with cell death [37]. Thus, in the present study, the WSS distribution in the scaffold geometry (Figure 13) is considered beneficial for promoting a higher proliferation rate and higher shear values that improve osteogenic differentiation, aligning with the study of Ali et al. [27]. It should be mentioned that a coupled fluid-structure interaction (FSI) simulation could better demonstrate the mutual interaction between fluid flow and structural deformations. However, we opted for separate FEA and CFD simulations, based on a careful evaluation of the available experimental data, the computational resources, and the different finite element solvers used for the bone scaffold simulation. We aim to expand this study in the future, with additional complex geometries, experimental data, and FSI simulations.

## 5. Conclusions

In this research, combined experimental and computational approaches were used to study and compare mechanical parameters, such as the apparent compression modulus, displacement, and von Mises stress, and fluidic parameters, such as fluid flow velocity, permeability, and fluid-induced WSS, in biohybrid nanocomposites scaffolds with specific rectangular-shaped geometry and porosity.

The overview of the results obtained from the combined experimental and computational mechanical analysis suggests the PLA scaffolds as the best candidates in load-bearing structures. Among the six different scaffolds, the PLA_LNG combines ultimate strength and high elasticity, while the PLA_MNG has the highest elasticity. Generally, the PLA scaffolds present higher endurance. On the other hand, the PCL scaffolds create a rubber-like mechanical behavior because their glass transition temperature (Tg) is lower than room temperature. These results are in line with typical bone requirements, while the fluid flow patterns, permeability, and WSS threshold values of the scaffold make it an optimal environment for supporting cellular growth and tissue regeneration.

## Figures and Tables

**Figure 1 biomedicines-12-00261-f001:**
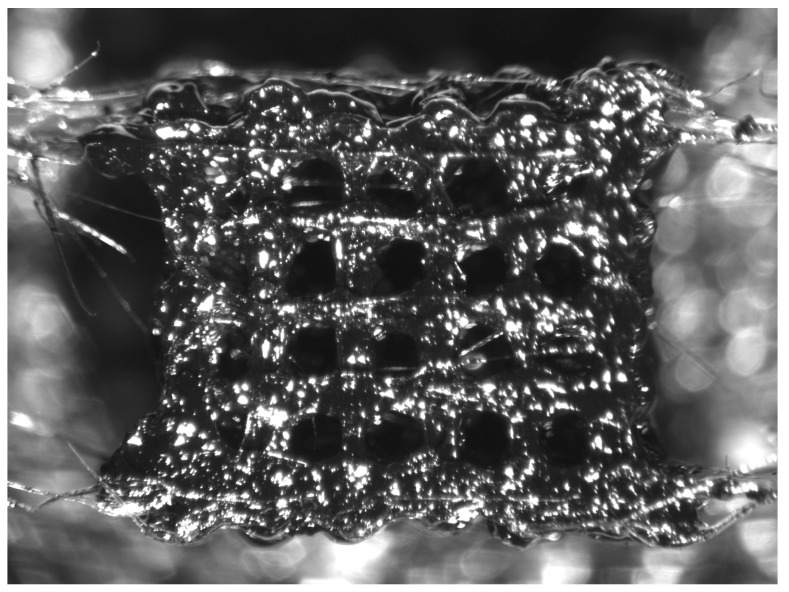
Reinforced PCL scaffold before compression.

**Figure 2 biomedicines-12-00261-f002:**
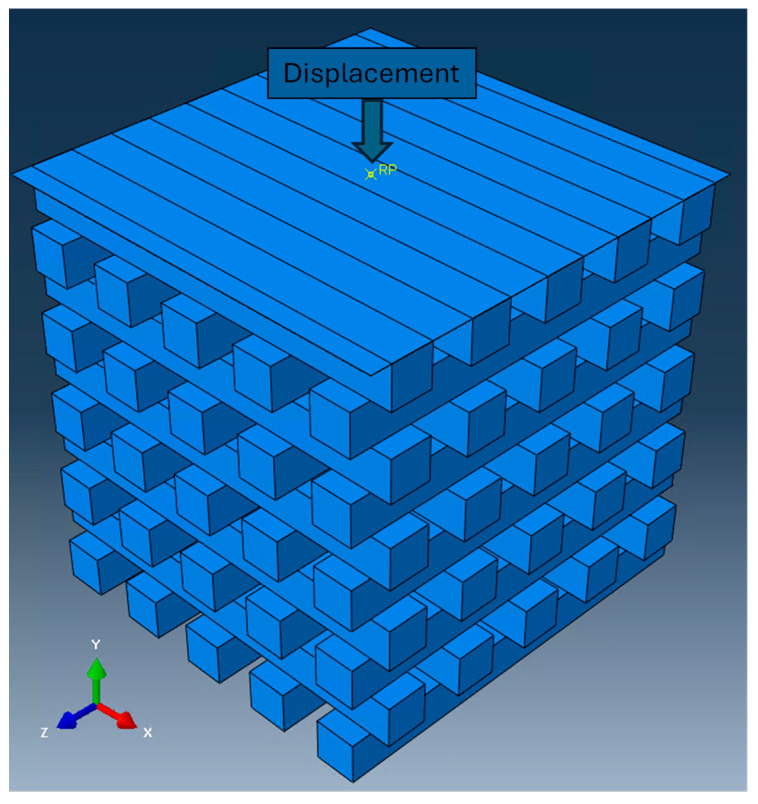
Scaffold design for finite element analysis.

**Figure 3 biomedicines-12-00261-f003:**
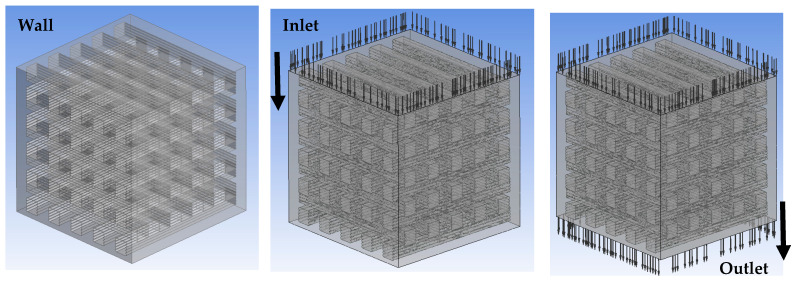
Boundary conditions of CFD analysis.

**Figure 4 biomedicines-12-00261-f004:**
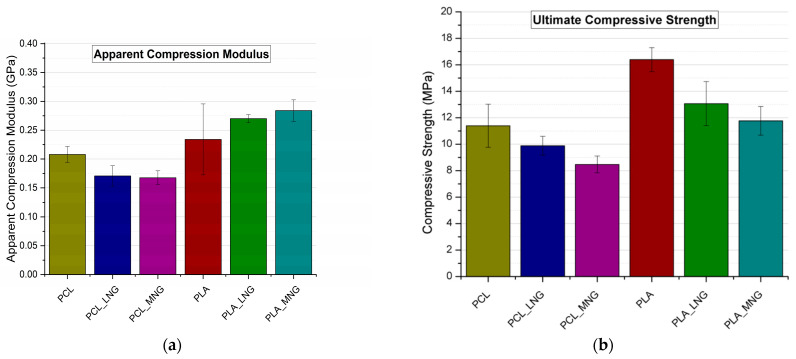
Mechanical properties of compressive experiments for all scaffolds: (**a**) experimental apparent compressive moduli; (**b**) experimental compressive strength.

**Figure 5 biomedicines-12-00261-f005:**
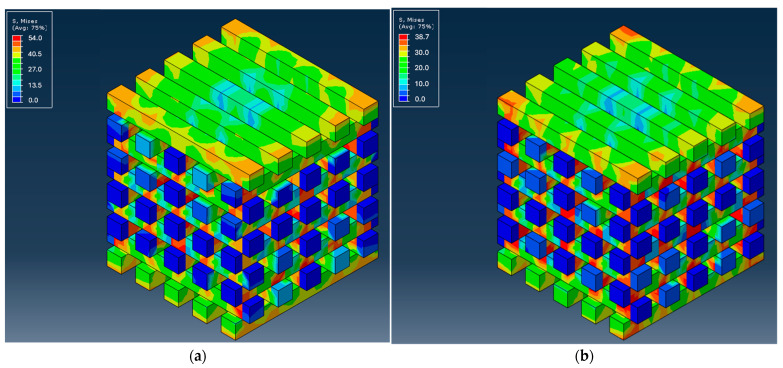
Von Mises stresses distribution for: (**a**) PLA scaffold; (**b**) PCL scaffold.

**Figure 6 biomedicines-12-00261-f006:**
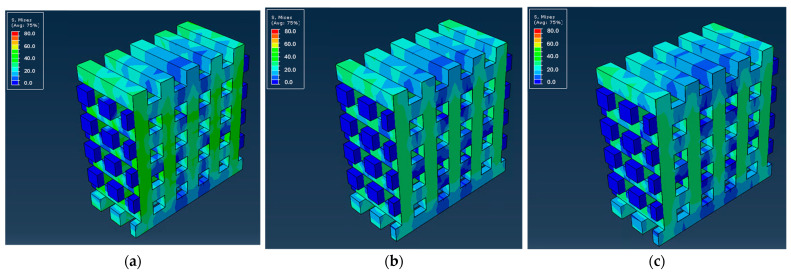
Slices of PCL scaffolds with the distributions of von Mises stresses inside the structure for: (**a**) PCL; (**b**) PCL_LNG; (**c**) PCL_MNG.

**Figure 7 biomedicines-12-00261-f007:**
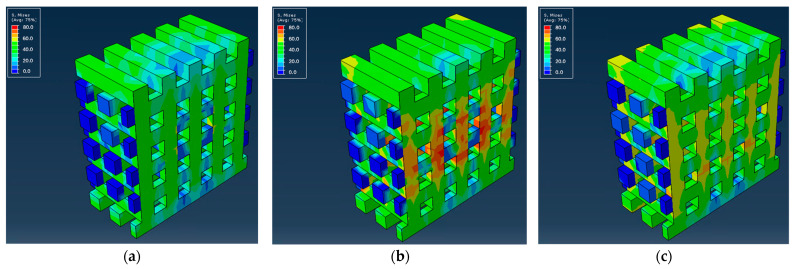
Slices of PLA scaffolds with the distributions of von Mises stresses inside the structure for: (**a**) PLA; (**b**) PLA_LNG; (**c**) PLA_MNG.

**Figure 8 biomedicines-12-00261-f008:**
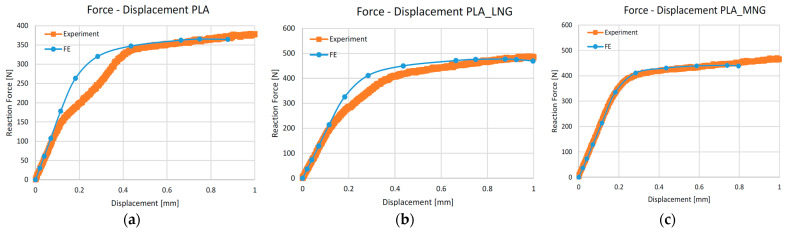
Force versus displacement curves for modeling and experiments for PLA scaffolds: (**a**) PLA; (**b**) PLA_LNG; (**c**) PLA_MNG.

**Figure 9 biomedicines-12-00261-f009:**
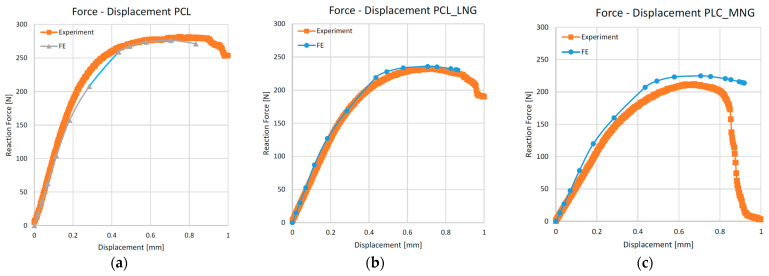
Force versus displacement curves for modeling and experiments for PLC scaffolds: (**a**) PLC; (**b**) PLC_LNG; (**c**) PLC_MNG.

**Figure 10 biomedicines-12-00261-f010:**
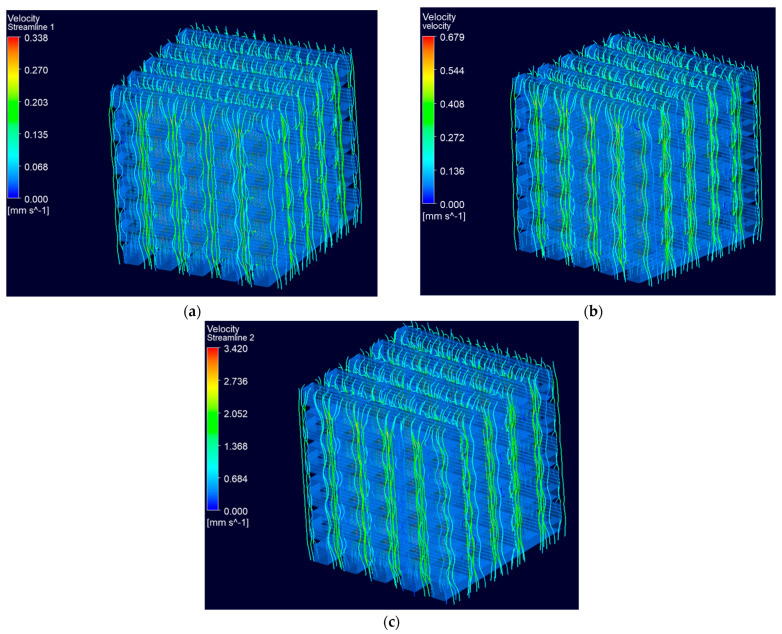
Velocity streamline distribution at (**a**) 0.05 mm/s, (**b**) 0.1 mm/s and (**c**) 0.5 mm/s, respectively.

**Figure 11 biomedicines-12-00261-f011:**
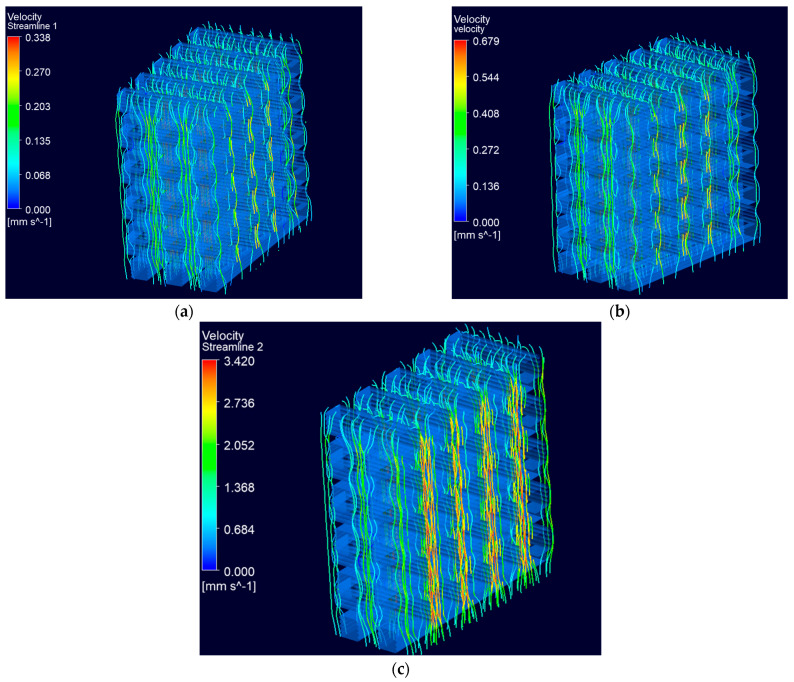
Cross-sectional views of the scaffold geometry with the distribution of velocity streamlines inside the structure at (**a**) 0.05 mm/s, (**b**) 0.1 mm/s and (**c**) 0.5 mm/s, respectively.

**Figure 12 biomedicines-12-00261-f012:**
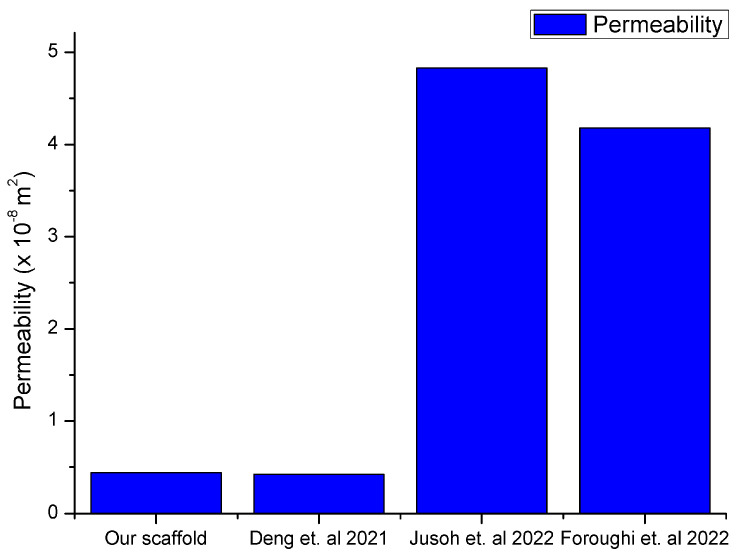
Comparison of the permeability results with data from the literature [12,26,28].

**Figure 13 biomedicines-12-00261-f013:**
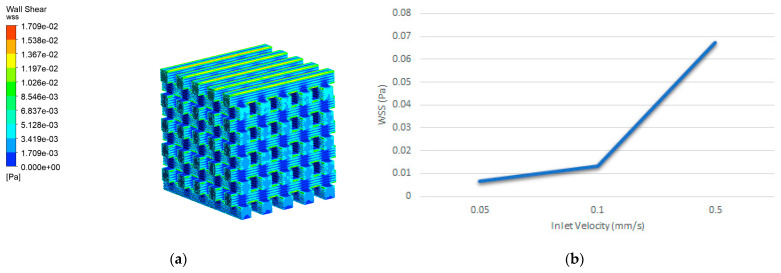
(**a**) WSS contour induced by an inlet velocity of 0.1 mm/s, (**b**) WSS distributions induced by an inlet velocity of 0.5, 0.1, and 0.05 mm/s in the scaffold.

**Table 1 biomedicines-12-00261-t001:** Experimental values for compression modulus for bulk/reference materials.

Materials	Compression Modulus (GPa)	Standard Deviation (GPa)
PCL	0.353	0.01061
PCL_LNG	0.361	0.00919
PCL_MNG	0.376	0.03764
PLA	3.122	0.12976
PLA_LNG	3.071	0.22762
PLA_MNG	3.305	0.18144

**Table 2 biomedicines-12-00261-t002:** Permeability results for the different inlet velocities.

Fluid Inlet Velocity [mm/s]	Permeability [m^2^]
0.5	4.315 × 10^−9^
0.1	4.412 × 10^−9^
0.05	4.413 × 10^−9^

## Data Availability

This study includes simulation and experimental data. Experimental data can be shared upon request made to the corresponding author.

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
