# Peer review of "A Combined Computational and Experimental Analysis of PLA and PCL Hybrid Nanocomposites 3D Printed Scaffolds for Bone Regeneration"

_biomedicines, 2024, doi:10.3390/biomedicines12020261_

Round 1

Reviewer 1 Report

Comments and Suggestions for Authors

The paper is focused on composite materials formed by PLA, PCL, chitosan and carbon nanotubes suitable as scaffolds for bone regeneration. The topic falls within the scope of the journal. The presentation and discussion of the results could be partly improved. On this basis, I recommend the publication after the following revisions:

-          Table 1. Please report the errors for the compressive modulus values.

-          I suggest to present the experimental stress vs strain curves.

-          Is it possible to determine other mechanical parameters (such as toughness) from the analysis of both computational and experimental compression curves?

-          Introduction could be updated by evidencing that the combination of chitosan with biopolymers and clay nanotubes has been explored for the fabrication of composite materials useful for tissue engineering [Journal of Materials Chemistry B, 2013,1, 2078-2089].

Comments on the Quality of English Language

Minor corrections are needed. 

Author Response

  • Table 1. Please report the errors for the compressive modulus values.

Response : Here we deposit the table with the errors. It was added also in the manuscript.

Compression Modulus

(GPa)

Standard deviation
(GPa)

Coefficient of Variation

(%)

PCL

0.3525

0.01061

3.009

PCL_LNG

0.3605

0.00919

2.55

PCL_MNG

0.376

0.03764

10.011

PLA

3.1215

0.12976

4.157

PLA_LNG

3.071

0.22762

7.412

PLA_MNG

3.305

0.18144

5.49

  • I suggest to present the experimental stress vs strain curves.

Response : The reason that we presented the load displacement curves is, that the load machine in the laboratory gives direct these curves and so, in order to have a direct validation

  • Is it possible to determine other mechanical parameters (such as toughness) from the analysis of both computational and experimental compression curves?

Response : Unfortunately the time duration of 1 week, to make the measurements and calculations, like toughness or more is very limited to export more mechanical parameters. But we can apply these ideas to the extension of this work. Form the other hand having the total load displacement curves which qualitatively is the same information with the stress strain, can

4) Introduction could be updated by evidencing that the combination of chitosan with biopolymers and clay nanotubes has been explored for the fabrication of composite materials useful for tissue engineering [Journal of Materials Chemistry B, 2013,1, 2078-2089].

Response : It was added.

Reviewer 2 Report

Comments and Suggestions for Authors

The manuscript "A Combined Computational and Experimental Analysis of Reinforced PLA and PCL Hybrid Nanocomposite Scaffolds for Bone Regeneration" reports on the preparation and characterization (experimental and theoretical) of nanocomposites scaffolds based on PLA, PCL, chitosan and MWCNTs intended for bone regeneration.

The composition of the paper is well presented and the suggested analyses: mechanical and fluid flow are the main features needed to integrate a scaffold in regenerative medicine.

I suggest minor corrections and some clarification regarding the advantages of the addition of MWCNTs in the composite materials.

Fig. 1 must be mention on which device is taken.

Table 1. Some comparisons from literature must be added.

In the Discussion section the authors must reinforce the relevance of the chosen composites as novel alternatives for bone regeneration.

Author Response

1)           Fig. 1 must be mention on which device is taken.

Response : Thank you for the useful comment. The microscope was Optika szm-Led2. It was added also in the main text.

2)           Table 1. Some comparisons from literature must be added

Response : The novelty of this study is the chemical system that we used to characterize it. This certain hybrid nanocomposites with PLAs or PCLs / MCNT/ Chitosan is extremely limited if not zero and for sure not in these fractions that we used. Although a comparison with the literature of familiar systems has been made in the in the main text already and it has been enrich in this version. Due to the limitation of words we cannot add another table.

3)           In the Discussion section the authors must reinforce the relevance of the chosen composites as novel alternatives for bone regeneration.

Response : The discussion has been enrich also and it is given the highlights and novelty of the current study.

Reviewer 3 Report

Comments and Suggestions for Authors

This article presents a thorough combined computational and experimental study evaluating the mechanical and fluid flow properties of 3D printed PLA and PCL scaffolds reinforced with chitosan functionalized multi-walled carbon nanotubes for bone regeneration applications. A comprehensive set of experiments characterized the bulk material and scaffold properties under compression through measurements of apparent compressive modulus and ultimate strength. Finite element analysis provided insights into stress distributions within the scaffolds and generally showed good agreement with experimental results. Computational fluid dynamics analysis yielded important permeability and wall shear stress data within the scaffold design. A systematic approach changing material formulations and geometry allowed the impact of these parameters to be evaluated. Overall, this work provides a rigorous multidisciplinary evaluation of these scaffold properties. However, some important questions still remain and need major revision:

1. How was the uniform dispersion of the carbon nanotube reinforcement ensured and assessed, both in bulk materials and printed scaffolds? Non-uniform dispersion could significantly impact the results. Clarification on the dispersion methodology and its effect is needed.

2. What challenges were encountered when 3D printing the scaffolds containing nanoscale reinforcement? Did the reinforcement impact printability or structural integrity at the resolutions achieved? More details on printing optimization would be useful.

3. Use the following reference to deepen the introduction and discussion. Unraveling of Advances in 3D-Printed Polymer-Based Bone Scaffolds. Toughening PVC with Biocompatible PCL Softeners for Supreme Mechanical Properties, Morphology, Shape Memory Effects, and FFF Printability. A New Strategy for Achieving Shape Memory Effects in 4D Printed Two-Layer Composite Structures.

4. For the discrepancy seen in Figure 8a-b between FEA and experimental force-displacement curves of certain scaffolds, the authors' perspective on potential causes should be provided to aid discussion.

5. The velocity streamlines presented in Figure 10 provide qualitative insights but could be enhanced for clarity.

6. Adding cross-sectional views of the velocity streamlines would significantly help readers better understand the three-dimensional flow distribution within the scaffold geometry.

7. Conducting a coupled fluid-structure interaction (FSI) analysis, rather than separate FEA and CFD simulations, could yield a more realistic simulation of how fluid flow influences mechanical deformation of the scaffold structure and vice versa. This approach may provide further valuable insights.

Comments on the Quality of English Language

**

Author Response

  1. How was the uniform dispersion of the carbon nanotube reinforcement ensured and assessed both in bulk materials and printed scaffolds? Non-uniform dispersion could significantly impact the results. Clarification on the dispersion methodology and its effect is needed.

  1. What challenges were encountered when 3D printing the scaffolds containing nanoscale reinforcement? Did the reinforcement impact printability or structural integrity at the resolutions achieved? More details on printing optimization would be useful.

Response: The first 2 comments will be addressed together as one due to the close relation of their content.

As stated in the manuscript

“Then the materials were extruded in the form of 3D-printing filament. The filament was once more pelletized, pulverized and then extruded in the form of filament or the form of bulk materials specimens. The pulverization process was repeated twice, in order to achieve a better dispersion of the filler material inside the PLA and PCL matrices.”

and also

“Generally, well-bonded fillers contribute to composites having notably higher strength, while weakly bonded particles can act as sources of inherent flaws, leading to a decrease in strength [29]. In the case of nano-scale reinforcement, nanoparticles tend to agglomerate more, creating crack-provoking flaws and reducing strength. This study observed that the addition of MWCNTs, especially in PCL loaded scaffolds led to nozzle clogging during the printing process, and the 3D-printed nano-composite exhibited increased brittleness [30] when compared to pure PCL scaffolds. The aforementioned factors, despite addressing them to the best of our abilities during manufacturing, could explain the overall reduction in ultimate compression strength observed.”

Since the filler concentration used was relatively high (5wt%), for nanoscale fillers, we already knew that some agglomerations were unavoidable. However, our goal was not to assess the reinforcing capabilities of the filler materials, hence we did not explore different fill rates; our fill rate was a constant 5wt%.

That is why we stated

“It should be noted that reinforcing the scaffold with different materials does not aim to significantly change the mechanical properties but to promote and improve cell proliferation and differentiation.”

Our goal was to achieve the maximum filler content, while the reinforced materials remained printable and did not negatively affect the initial mechanical properties, which we achieved. In order to improve our manuscript and further illustrate the aforementioned statements, in accordance with the reviewer’s comments, the following alterations to the original manuscript have been made:

The sentence was added,

 “The goal was to achieve the maximum filler content, while the reinforced materials remained printable and did not negatively affect the initial mechanical properties. It was found in preliminary work that the weight fraction of 5% was the maximum attained weight filler fraction that the materials remained printable”

Also   “The pulverization process was repeated twice, in order to break any existing large agglomerations and achieve a better dispersion of the filler material inside the PLA and PCL matrices.”

(The phrase in bold was added.)

Finally, in the Discussion section we added:

“This was to be expected, since our goal was to achieve the maximum filler content, while the reinforced materials remained printable and did not negatively affect the initial mechanical properties in order to promote and improve cell proliferation and differentiation.”

  1. Use the following reference to deepen the introduction and discussion. Unraveling of Advances in 3D-Printed Polymer-Based Bone Scaffolds. Toughening PVC with Biocompatible PCL Softeners for Supreme Mechanical Properties, Morphology, Shape Memory Effects, and FFF Printability. A New Strategy for Achieving Shape Memory Effects in 4D Printed Two-Layer Composite Structures.

It was added. This article seems very interesting indeed.

  1. For the discrepancy seen in Figure 8a-b between FEA and experimental force-displacement curves of certain scaffolds, the authors' perspective on potential causes should be provided to aid discussion.

Response: Very useful comment. In this direction we added that:
“This may be due to impurities that have been induced during manufacturing process. As we can see these two systems in contrast with the PLA_MNG, are experiencing a smoother transition to plastic region. In this direction impurities can work in such a way that they intensify the transition between the two regions.”

  1. The velocity streamlines presented in Figure 10 provide qualitative insights but could be enhanced for clarity.

  1. Adding cross-sectional views of the velocity streamlines would significantly help readers better understand the three-dimensional flow distribution within the scaffold geometry.

Response: The comments 5 and 6 will be addressed together as they are referred to the same content, i.e. the velocity streamlines.

Thank you for pointing these out. We agree with these comments. We have, accordingly, replaced Figure 10 with an enhanced quality figure regarding the velocity streamline distribution inside the scaffold at different inlet velocities.

For a better comprehension of the flow distribution within the scaffold geometry we provided cross-sectional views of the velocity streamlines, as you requested (Figure 11).

we added the statement:

“Figure 11 demonstrates the distribution of velocity streamlines at the middle cross-section for the scaffold geometry.”

As stated in the manuscript:

“The higher velocity in the struts, as depicted in Figure 11c, indicates the increase in the rate which is favorable for the absorption of cells and nutrients on the inner surface of the scaffold.”

*We changed Figure 10c to Figure 11c (in bold).

We added a further explanation:

“This accelerated velocity creates an environment conducive to more efficient cellular interactions and improved nutrient uptake, ultimately contributing to the overall effectiveness of the scaffold in supporting biological processes [22].”

  1. Conducting a coupled fluid-structure interaction (FSI) analysis, rather than separate FEA and CFD simulations, could yield a more realistic simulation of how fluid flow influences mechanical deformation of the scaffold structure and vice versa. This approach may provide further valuable insights

Response: Thank you for this suggestion. It would have been interesting to investigate the interaction between fluid flow and mechanical deformation. However, in our case FEA and CFD are separate, well-established and validated for bone scaffold simulations conducted by utilizing different finite element solvers (Abaqus and Ansys software, respectively) [1, 2] that provide more accurate and reliable results (Par. 3.2.1, 3.2.2). To that end, the major limitation we encountered in the specific work, is that the available experimental data primarily focuses on mechanical properties (Table 1, Figure 4), thus, the coupled effects of fluid flow and mechanical deformation haven’t been addressed in the bone scaffold and it might be challenging and would introduce additional complexities and potential sources of error to justify the inclusion of a coupled FSI analysis [3]. Last but not least, FSI simulations are notoriously computationally more intensive than CFD [4], particularly when applied to intricate scaffold structures or complex geometries such as the proposed one. Therefore, based on our constrained computational resources, conducting discrete FEA and CFD simulations seems more practical and feasible for the purpose of the specific study.

Overall, even though the FSI simulations are powerful for demonstrating the mutual interaction between fluid and structure, our decision to choose separate FEA and CFD simulations is based on a careful evaluation of the available experimental data, the computational resources, and the different finite element solvers used for the bone scaffold simulation. It's essential to strike a balance between accuracy and computational efficiency based on the goals of the study.

References

[1] Omar AM, Hassan MH, Daskalakis E, Ates G, Bright CJ, Xu Z, Powell EJ, Mirihanage W, Bartolo PJDS. Geometry-Based Computational Fluid Dynamic Model for Predicting the Biological Behavior of Bone Tissue Engineering Scaffolds. J Funct Biomater. 2022 Jul 27;13(3):104.

[2] Hassan CR, Qin YX, Komatsu DE, Uddin SMZ. Utilization of Finite Element Analysis for Articular Cartilage Tissue Engineering. Materials (Basel). 2019 Oct 12;12(20):3331.

[3] Zhao F, van Rietbergen B, Ito K, Hofmann S. Fluid flow-induced cell stimulation in bone tissue engineering changes due to interstitial tissue formation in vitro. Int J Numer Method Biomed Eng. 2020 Jun;36(6):e3342. doi: 10.1002/cnm.3342. Epub 2020 May 6. PMID: 32323478; PMCID: PMC7388075.

[4] Xu, L., et al. (2019). "Computational methods applied to analyze the hemodynamic effects of flow-diverter devices in the treatment of cerebral aneurysms: Current status and future directions." Medicine in Novel Technology and Devices 3: 100018.

Round 2

Reviewer 1 Report

Comments and Suggestions for Authors

The paper was improved according to the reviewers' suggestions. I recommend its publication in the current form.

Author Response

Thank you very much!

Reviewer 3 Report

Comments and Suggestions for Authors

The format of the references used is not consistent with the journal. In addition, the provided answers have not been applied in the text of the article, especially comments 1 and 3.

Author Response

Reviewer

The format of the references used is not consistent with the journal.

RESPONSE: The format of the references was changed.

Reviewer

In addition, the provided answers have not been applied in the text of the article, especially comments 1 and 3.

RESPONSE:

  • We kindly ask you to observe carefully the text of the article. The provided answers of comment 1 had been applied in the text from the previews version.

You can see the lines   

149-153 for “Then the materials were extruded in the form of 3D-printing filament. The filament was once more pelletized, pulverized and then extruded in the form of filament or the form of bulk materials specimens. The pulverization process was repeated twice, in order to achieve a better dispersion of the filler material inside the PLA and PCL matrices.”

411-423, “Generally, well-bonded fillers contribute to composites having notably higher strength, while weakly bonded particles can act as sources of inherent flaws, leading to a decrease in strength [32] . In the case of nano-scale reinforcement, nanofillers tend to agglomerate more, creating crack-provoking flaws and reducing strength [33,34]. This study observed that the addition of MWCNTs, especially in PCL loaded scaffolds led to nozzle clogging during the printing process, and the 3D-printed nano-composite exhibited increased brittleness when compared to pure PCL scaffolds. This was to be expected, since our goal was to achieve the maximum filler content, while the reinforced materials remained printable and did not negatively affect the initial mechanical properties in order to promote and improve cell proliferation and differentiation. The aforementioned factors, despite addressing them to the best of our abilities during manufacturing, could explain the overall

reduction in ultimate compression strength observed.”

274-276, “It should be noted that reinforcing the scaffold with different materials does not aim to significantly change the mechanical properties but to promote and improve cell proliferation and differentiation.”

 141-145, The goal was to achieve the maximum filler content, while the reinforced materials remained printable and did not negatively affect the initial mechanical properties. It was found in preliminary work that the weight fraction of 5% was the maximum attained weight filler fraction that the materials remained printable.

152,  to break any existing large agglomerations

418-421: “This was to be expected, since our goal was to achieve the maximum filler content, while the reinforced materials remained printable and did not negatively affect the initial mechanical properties in order to promote and improve cell proliferation and differentiation.”

  • The reference paper of comment 3
    Unraveling of Advances in 3D-Printed Polymer-Based Bone Scaffolds. Toughening PVC with Biocompatible PCL Softeners for Supreme Mechanical Properties, Morphology, Shape Memory Effects, and FFF Printability. A New Strategy for Achieving Shape Memory Effects in 4D Printed Two-Layer Composite Structures”

also it had been already added from the previews version. You can see the ref.13

Round 3

Reviewer 3 Report

Comments and Suggestions for Authors

Accept.